# Lake O'Hara alpine hydrological observatory: Hydrological and meteorological dataset, 2004-2017

Jesse He and Masaki Hayashi

Department of Geoscience, University of Calgary, Alberta, Canada

*Correspondence to*: Jesse He (jehe@ucalgary.ca)

**Abstract.** The Lake O'Hara watershed in the Canadian Rockies has been the site of several hydrological investigations. It has been instrumented to a degree uncommon for many alpine study watersheds. Air temperature, relative humidity, wind, precipitation, radiation, and snow depth are measured at two meteorological stations near Lake O'Hara and in the higher elevation Opabin Plateau. Water levels at Lake O'Hara, Opabin Lake, and several stream gauging stations are recorded with

pressure transducers and validated against manual measurements. Stage-discharge rating curves were determined at gauging stations and used to calculate discharge from stream stage. The database includes additional data such as water chemistry (temperature, electrical conductivity, and stable isotope abundance) and snow survey (snow depth and density) for select years, as well as geospatial data (elevation and land cover). This dataset will be useful for future study of alpine regions, where substantial and long-term hydrological datasets are scarce due to difficult field conditions. The dataset can be accessed at:

https://doi.org/10.20383/101.035

## 1 Introduction

Mountains are an important source of water for downstream regions (Viviroli et al., 2007). The hydrology of mountains in mid- and high-latitude regions is dominated by the storage of water in the form of snowpack and glaciers, which provide melt water to headwater streams during the melt season (Barnett et al., 2005). Climate warming can influence these processes

through changes in the timing of snow accumulation and melt, the transition towards more rain and less snow, and depletion of glaciers and mountain permafrost (Bales et al., 2006). While it is straightforward to understand the effects of these changes on the timing and magnitude of spring freshet, their effects on the flow during low-flow periods is uncertain because groundwater can store and release melt waters, thereby buffering the effects of warming (Tague and Grant, 2009). However, groundwater processes in alpine headwaters are not well understood due to the lack of long-term, field-based studies examining

the interaction of surface water and groundwater in alpine zones. To address this gap in knowledge, an alpine hydrological observatory was established in the watershed of Lake O'Hara in the Canadian Rocky Mountains (see Study Site below) in 2004. The observatory has supported a series of studies designed to identify important alpine aquifer units and characterize their hydrogeological functions (Roy and Hayashi, 2009; Langston et al., 2011; Muir et al., 2011; Hood and Hayashi, 2015). The observatory was integrated into a larger hydrological monitoring network under the Changing Cold Regions Network

(CCRN) in 2013 (DeBeer et al., 2016). The unique dataset collected to date are available to the public through the CCRN data server. The objective of this paper is to describe the physiographical characteristics of the Lake O'Hara watershed, the history of the hydrological observatory, and the methods of data collection and processing for the potential users of the dataset.

## 2 Site description

The Lake O'Hara watershed in Yoho National Park has an area of 14 km$^2$ and ranges in elevation from 1996 to 3440 meters
above sea level (m.a.s.l.) (Figure 1). The site is accessible via an 11-km dirt road up to Lake O'Hara. Raw mean annual precipitation measured within the watershed at the Opabin automatic weather station (OPAWS) was 1021 mm during 2005-2017. After adjustment for noise and wind induced undercatch (Kochendorfer et al., 2017), corrected annual precipitation at OPAWS was 1113 mm. Mean monthly air temperatures at OPAWS range from -9.0 (January) to 10.2 $^o$C (July).

Bedrock throughout the watershed consists mainly of quartzite of the Cambrian Gog Group. At higher elevations, carbonate of the Cathedral, Mt. Whyte, Stephen and Eldon Formations may be found capping mountains (Lickorish and Simony, 1995; Price et al., 1980). Bedrock is typically exposed at higher elevations as part of ridges which ring the watershed (Figure 2a). Overburden deposits (e.g. talus and moraine) are present throughout the region, and are found in association with steeper bedrock slopes and small glaciers (Figure 2b and 2c). These deposits play an important role in groundwater and surface water
exchange throughout the watershed (Langston et al., 2011; Muir et al., 2011; Roy et al., 2009). For example, 70-80% of flow in Upper Opabin Creek is provided from a large talus-moraine complex present in the Opabin plateau (Hood and Hayashi, 2015). Surface water bodies such as Opabin Lake, Hungabee Lake and the eponymous Lake O'Hara are found at lower elevations in the watershed. Previous study has indicated active interaction between lakes and groundwater (Roy and Hayashi, 2008). Groundwater input to Lake O'Hara was estimated to equal 35-74% of surface outflow from the lake in 2005 (Hood et
al., 2006). Alpine meadows (Figure 2d) are generally found in close association with surface water bodies and are important sites for hydrological and ecological processes (McClymont et al., 2010).

Compared to other high-elevation lakes in the region, the Lake O'Hara hydrological observatory is easily accessible by a road, yet relatively undisturbed (road access is only allowed for a small number of authorized vehicles), making it ideal for the study
of alpine hydrological processes. Monitoring at Lake O'Hara began in 2004 with the installation of OHAWS, OPAWS and several stream gauging and water level monitoring stations (Figure 1). Some change has been made to the number and locations of gauging stations. Since 2009, the Gorge gauging station has replaced the Opabin East and West stations for the measurement of discharge in the lower reaches of Opabin Creek. In Oesa Creek, measurements of discharge have been taken at the Oesa Falls (2005-2006, 2008 and 2014-2017) and Lefroy Oesa (2013-2014) gauging stations over different years. Water levels at
Lake O'Hara and Opabin Lake have been measured continuously during the ice-free season since 2004. Additional equipment have been installed and removed as part of previous studies conducted within the watershed. Study-specific data include

measurements from temporary sensors and stream gauging stations as well as extensive geophysical, geochemical, and snow survey data. The CCRN dataset only include consistent measurements from permanent weather stations, lake stilling wells and stream gauging stations (Figure 1) over the years of 2004-2017. Some chemistry and snow survey measurements are also included for select years.

## 3 Methods

### 3.1 Meteorological data

Two permanent weather stations are present within the Lake O'Hara watershed (Figure 1). OHAWS is located 500 m northwest of Lake O'Hara at an elevation of 2024 m.a.s.l. OPAWS is located on the Opabin plateau at 2230 m.a.s.l.

Both stations are equipped with air temperature and relative humidity sensors (Vaisala, HMP35) which are kept within solar radiation shields to mitigate the influence of direct sunlight on measurements. Wind speed and direction are measured with vane anemometers (RM Young, 05103). A four component radiometer (Kipp & Zonen, CNR-1) measures incoming and outgoing fluxes of longwave and shortwave radiation at OPAWS. At OHAWS, a net radiometer (Kip & Zonen, NR-Lite) measures net radiation flux. Both stations have an ultrasonic snow depth sensor (Campbell Science, SR50) measuring the distance between the sensor and ground/snow level below. Temperature-compensated snow depth data are calculated from raw SR50 data and the sensor height. All aforementioned sensors are mounted on a tripod at both stations. Recent sensor heights are listed in Table 1. Measurements are taken every minute, and average values are recorded at an interval of 60 and 30 minutes at OHAWS and OPAWS, respectively.

Tipping bucket rain gauges (Hydrological Services, CS700) and weighing cumulative precipitation gauges (Geonor, T200B) are installed at both stations. Both types of precipitation gauge are suitable for measuring liquid precipitation, however, only the weighing gauge is capable of measuring solid precipitation. Tipping bucket gauges are placed on the ground, away from the main weather station tripod. Weighing gauges are mounted on a freestanding base and are equipped with an Alter wind shield to reduce wind-related snow undercatch. Precipitation and relative humidity sensors are calibrated yearly in order to ensure that they continue to accurately record measurements. However, relative humidity measurements fail to reach 100% in several years, possibly due to the instrumental accuracy (2% reported by the manufacturer) affecting the two-point calibration procedure using the saturated chemical solutions with known equilibrium humidity. In particular, the higher end of the data was substantially under-calibrated during 2011-2013 with maximum values reaching only 91%. However, we have opted to keep the raw values in the database rather than making a subjective correction and introducing another degree of uncertainty.

Meteorological measurements in the dataset are presented largely in raw form; some effort has been made to remove erroneous data due to sensor malfunction or maintenance in the raw dataset, but the data have largely been spared from other corrections.

Precipitation data are available in both raw and corrected form. The corrected precipitation dataset has been adjusted for noise and wind induced undercatch (Kochendorfer et al., 2017). Undercatch-corrected precipitation was 8.6% and 0.3% higher than raw precipitation at OPAWs and OHAWS respectively. From 2011 to 2016, snow has been observed to accumulate at the top of the OHAWS weighing precipitation gauge and persist over the span of several weeks in the wintertime. This accumulation effectively blocks the opening of the gauge (i.e. snow capping), resulting in a period of time where no precipitation is registered. We attempt to approximate this missing data at OHAWS from measurements recorded at OPAWS in the corrected dataset. Comparison of noise and undercatch-corrected daily precipitation between the two weather stations indicates a linear correlation (Figure 3). Using the trend line from Figure 3, precipitation at OHAWS was approximated and used to fill in gaps during periods of snow capping. Table S1 in the supplement lists the date ranges where precipitation was approximated, and total approximated precipitation.

### 3.2 Stream discharge

Gauging stations are located at the outlets of Lake O'Hara and Hungabee Lake and along reaches of several creeks (Figure 1). To establish the discharge-stage rating curve, biweekly manual discharge measurements are taken at each station over field seasons which typically last from early June to late September. Average stream flow velocity and water depth are measured in 10 or 25 cm wide segments across the width of a stream with a propeller flow meter (Global Water, FP101), and discharge is calculated by the area-velocity method (Dingman, 2002, p609).

Pressure transducers (In-Situ, Minitroll; In-Situ, LevelTroll; Solinst, Levelogger) are installed in stilling wells at each stream gauging station and automatically record water level and temperature in fifteen minute intervals. To verify the transducer data, water levels are measured manually from a reference (staff gauge or stilling well) with discharge measurements. A power function is used to define the rating curve, which is calibrated each year to find a coefficient and exponent which minimize error between measured and computed discharge. With the rating curve, a near-continuous record of stream discharge can be calculated from the transducer data. Hourly and half-hourly averaged discharge is included within the dataset. Discharge is only measured from spring (typically early June) to fall (typically late September) of each year due to winter freeze up in stream channels.

### 3.3 Lake water level

Pressure transducers housed in stilling wells are used at water level monitoring stations in Lake O'Hara and Opabin Lake (Figure 1). Transducer water levels are compared against manual measurements of water level taken during field visits. At Lake O'Hara, manual measurements are taken as the distance from the top of the stilling well casing to the lake water level. At Opabin Lake, manual measurements are taken as the distance from a rock bolt (securing the stilling well) to the lake water level. Within the dataset, both raw water level (depth of water above the transducer) and corrected water level are included. Corrected water level is normalized against the benchmark and is calculated as the distance between the benchmark and the

water level. Therefore, corrected water level increases when lake water level (and raw water level) drops. The same is true in reverse. Negative values of corrected water level indicate periods when lake water level is above the datum. Corrected water levels are not available in several years due to inconsistent manual measurement and recording of lake water levels (Table S2). Water levels are only available from spring to fall of each year due to freeze up.

**3.4 Snow surveys**

Snow depth and density are measured annually in mid-April to capture the amount of peak accumulation. The extent of these snow surveys has varied over the years from a handful of transects nearby OHAWS and OPAWS to hundreds of measurements spanning the entire Opabin plateau. Snow surveys are typically conducted by laying a measurement chain along the survey transect and measuring snow depth at a fixed interval with a probe or ruler. Snow density measurements are taken along the

survey transect or from snowpits (see Hood and Hayashi, 2015 for details). Handheld Global Positioning System devices are used to locate depth and density measurement points. Snow survey data from 2006 to 2017 are included within the dataset. Before 2012, measurements near OHAWS were made inconsistently, but in more recent years, data from both OPAWS and OHAWS are available (Table S2 in the supplement).

**3.5 Water sample collection and analysis**

Stream water and rain samples are collected during biweekly site visits. Stream water is sampled at most gauging stations (Figure 1). During sampling, water is filtered in the field with 0.45 µm disposable filters and stored in pre-rinsed polyethylene bottles. Electrical conductivity and temperature of stream water are measured during collection with handheld meters (VWR, 2052-B; Omega Engineering, HH-25TC). Rain samples are collected from samplers deployed near both weather stations and Opabin Lake. Depth-integrated snow samples were collected from snow pits during snow survey campaigns in 2015 and 2016,

and sub-samples of entire melt water were kept for isotope analysis. All water samples are stored at 4 °C until analysis. Oxygen-18 and deuterium isotope abundances are measured in all collected samples using an off-axis integrated-cavity spectrometer (Los Gatos Research, DLT-100). This dataset includes chemical and stable isotope data collected from 2004-2008, 2013, and 2015-2016.

**3.6 Spatial data**

The dataset includes a 2-m resolution digital elevation model (DEM) of the Lake O'Hara region derived from the light detecting and ranging (LiDAR) data (Hopkinson et al., 2009). Some difference in elevation has been found between the LiDAR DEM provided and older, lower resolution topographic maps of the region, possibly resulting from inconsistency in the elevation datum used during the LiDAR survey. However, the LiDAR DEM still represents topography within the watershed accurately. Also included are several ArcMap (ESRI, 2014) shapefiles which classify land cover throughout the watershed (Figure 1).

Land cover was classified manually from an aerial photo taken in August 2006.

## 4 Data examples

Figure 4 show the annual time series of hydro-meteorological parameters indicating the inter-annual variability of hydrological fluxes during hydrological years 2006−2017. Note that the hydrological year (HY) in these plots are defined as October 1 – September 30; e.g., HY2006 starts on October 1, 2005. Winter is defined as October 14 – April 30, because the average start
date of snow accumulation is October 14, and the accumulation normally peaks in late April. Annual total precipitation at Opabin AWS ranged between 920 and 1394 mm with a mean of 1113 mm, while winter precipitation ranged between 475 and 785 mm with a mean of 612 mm (Figure 4a). The ratio of winter to total precipitation ranged between 0.46 and 0.65. Annual mean and winter mean air temperature ranged from -2.4 to -0.1 ºC and from -8.9 to -5.4 ºC, respectively (Figure 4b). There was no noticeable trend in any of these parameters.


The timing of snow accumulation and melt are believed to respond sensitively to the climate warming (e.g. Barnett et al., 2005). We define the first day of accumulation at Opabin AWS as the day when snowpack starts to persist continuously, and the first day of complete melt as the day when the snow-depth sensor indicates no snow. These dates varied widely with the first day of melt from May 29 to July 3 (Figure 4c) and the first day of accumulation from September 28 to October 26 (Figure
4b), but there was no noticeable trend.

The total discharge (i.e. watershed runoff) during June-September measured at Lake O'Hara outlet had a large variability (Figure 4e) and was positively correlated with total annual precipitation ($r$, correlation coefficient = 0.78; $p$, significance level = 0.003) and winter precipitation ($r = 0.65$, $p = 0.02$), as expected. Previous studies in the region have shown that late-summer
flow in alpine streams is predominantly sourced by groundwater (e.g. Hood and Hayashi, 2015; Harrington et al., 2018). However, discharge in September (not plotted) was not correlated to either total annual precipitation ($r = 0.01$, $p = 0.75$) or Jun-Aug precipitation ($r = 0.003$, $p = 0.86$) suggesting that the groundwater discharge rate may be controlled by factors other than precipitation, such as the storage capacity or transmissivity of aquifers.

## 5 Data availability

The Lake O'Hara dataset is available from a public database operated by the CCRN (http://giws.usask.ca/KistersWeb/main.php). The dataset is also stored at the Federated Research Data Repository (FRDR), and can be accessed from the FRDR at: https://doi.org/10.20383/101.035

## 6 Final remarks

The data from the Lake O'Hara hydrological observatory have contributed significantly to our understanding of groundwater
processes in alpine environments. The long-term dataset can be used to examine the inter-annual variability of hydrological

fluxes and the timing of snow accumulation and melt, and their long-term trends. The unique dataset will be valuable to alpine hydrological research communities for various purposes such as inter-site comparison of hydrological processes or hydrological model testing.

## Competing interests

The authors declare that they have no conflict of interest.

## Acknowledgment

The field program at Lake O'Hara was assisted by numerous field assistants, who are too many to name. We especially thank the graduate students and post-doctoral fellows who conducted hydrological research projects: Jaime Hood, James Roy, Danika Muir, Greg Langston, and Andrius Paznekas. We also thank field technicians who took responsibility for data collection and

quality control: Mike Toews, Jacqueline Schmidt, Nathan Green, Jackie Randall, Kate Forbes, Krystal Chin, Shelby Snow, and Brandon Hill. Chris Hopkinson and the Canadian Consortium for Lidar Environmental Application Research provided LiDAR data for the DEM, Parks Canada and Lake O'Hara Lodge provided logistical supports, and Branko Zdravkovic and Amber Peterson assisted with data transfer and archive. The program has been funded by Natural Sciences and Engineering Research Council (Discovery Grant, CCRN), Biogeoscience Institute of the University of Calgary, Alberta Ingenuity Centre

for Water Research, Canadian Foundation for Climate and Atmospheric Sciences (IP3 Network), Canada Foundation for Innovation, Canada Research Chair program, and Environment Canada. Constructive comments by Ignacio López-Moreno and an anonymous reviewer contributed to improved clarity of the paper.

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

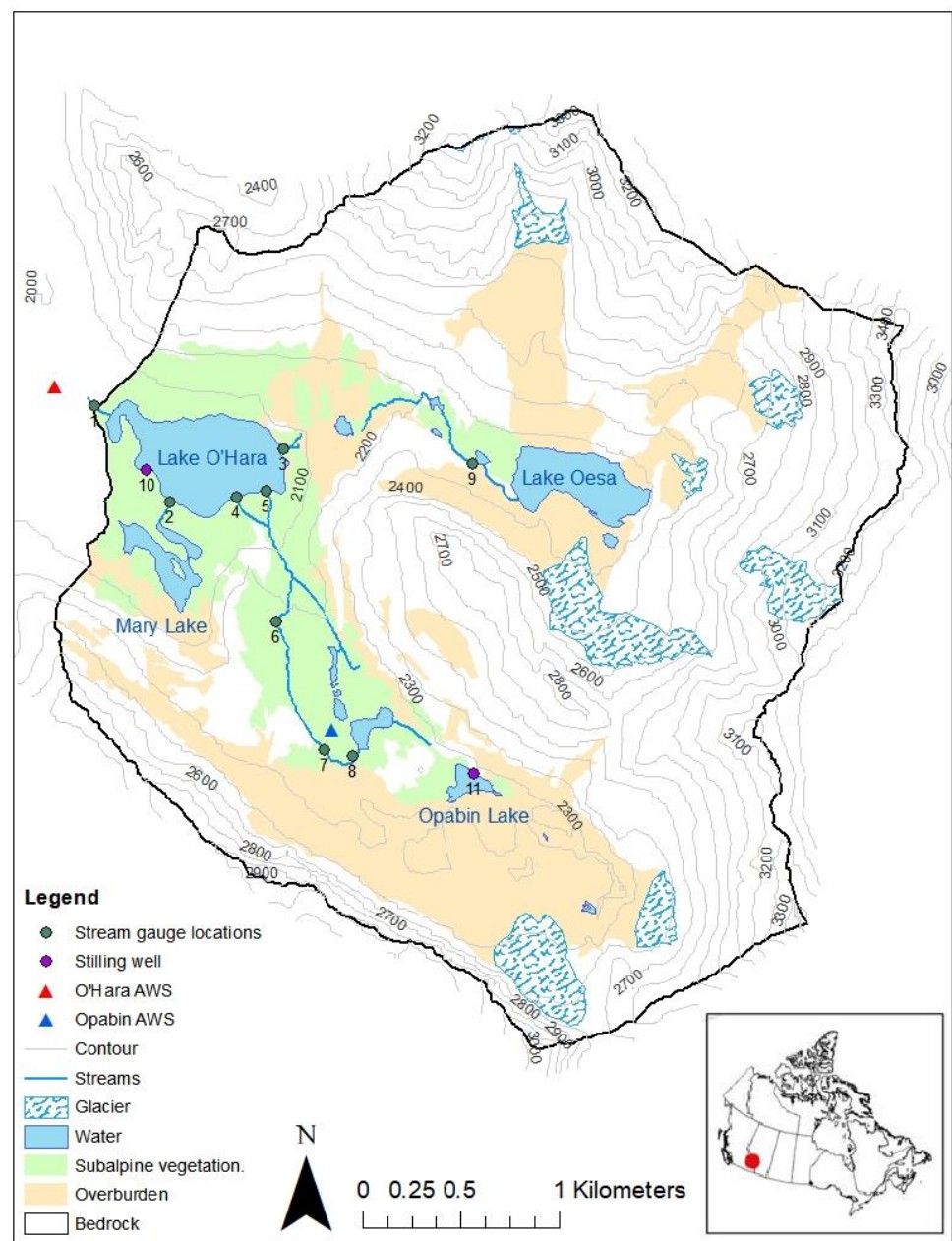

Figure 1. Topographic map of the Lake O'Hara watershed indicating major landcover units and locations of installed equipment. Stream gauging stations are: (1) O'Hara outlet, (2) Mary, (3) Oesa Falls, (4) West Opabin, (5) East Opabin, (6) Gorge, (7) Upper Opabin, (8) Hungabee, (9) Lefroy Oesa. Stilling wells are: (10) Lake O'Hara, (11) Opabin Lake. See Table
S3 for station coordinates. Landcover delineation was based on 2006 aerial photography; current glacial extents are smaller than indicated on the map.

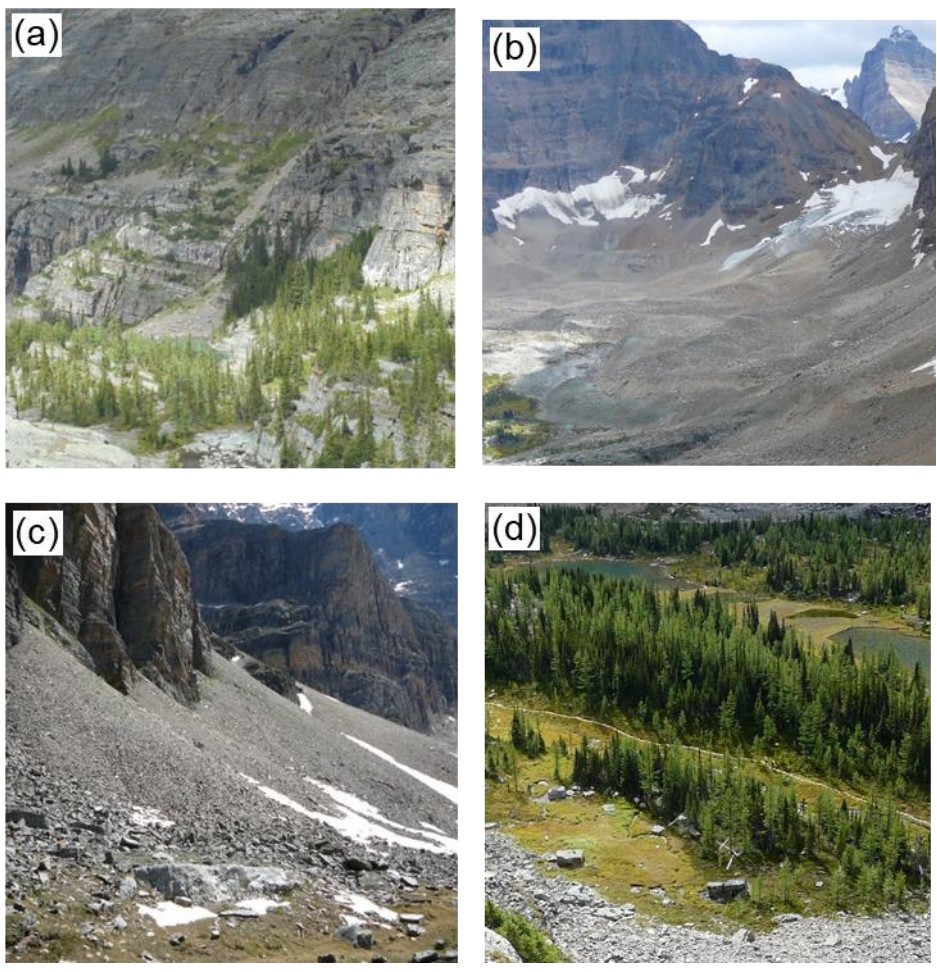

Figure 2. Major hydrogeological response units in the Lake O'Hara watershed. (a) Exposed bedrock. (b) Proglacial moraine. (c) Talus. (d) Alpine meadow.


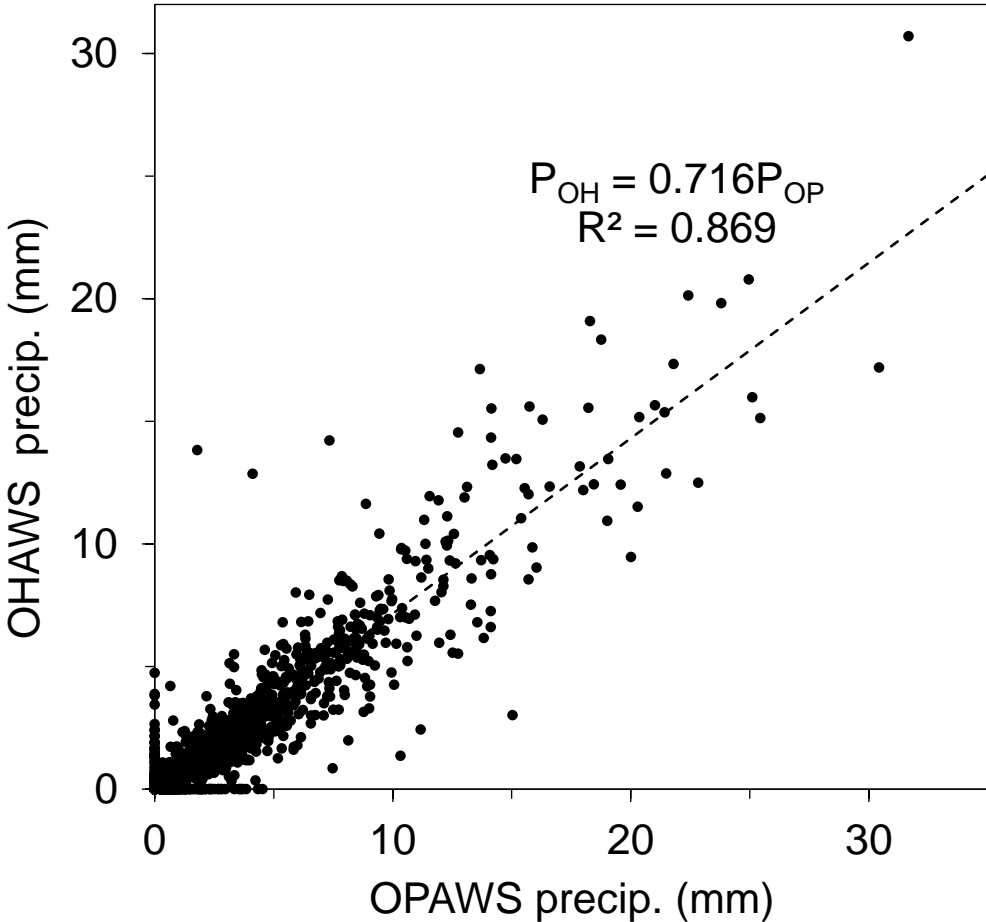

Figure 3. Relationship between daily winter precipitation at OHAWS ($P_{OH}$) and OPAWS ($P_{OP}$). Daily precipitation was corrected for instrument noise and wind-induced undercatch. Plotted daily precipitation was sampled from the entire monitoring period (2004 – 2017) for days between Dec 1 and Apr 30 where both OPAWS and OHAWS were recording and correctly functioning the entire day. $R^2$ is the coefficient of determination.

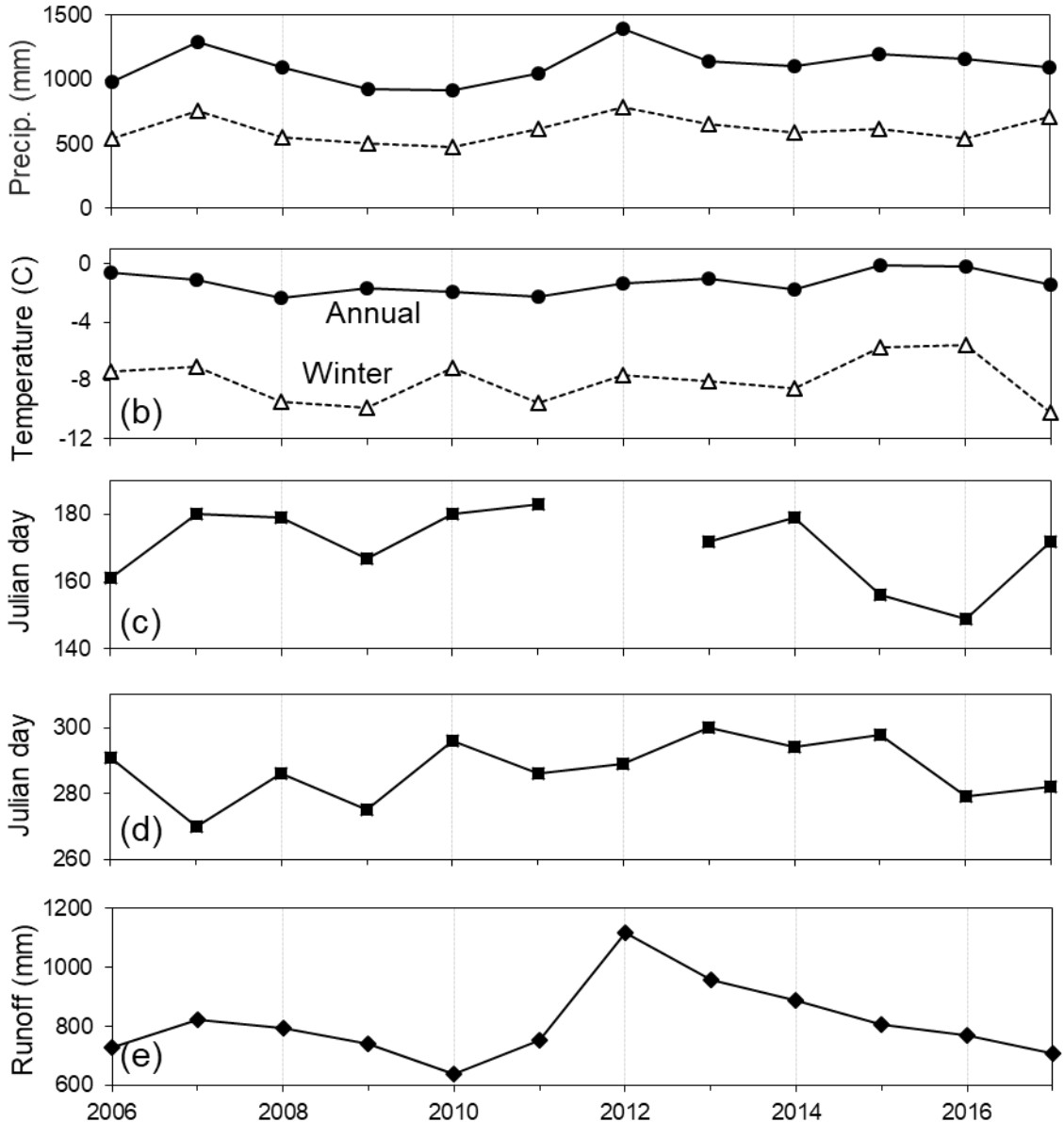

Figure 4. (a) Annual total and winter (October 14-April 30) precipitation at OPAWS. (b) Mean annual and winter (October 14-April 30) air temperature at OPAWS. (c) First day of complete melt at OPAWS. Data is missing in 2012 due to sensor malfunction. (d) First day of snow accumulation at OPAWS. (e) June-September total runoff at Lake O'Hara outlet.

Table 1. Current sensor height measurements at OHAWS and OPAWS. A more complete record of sensor height changes over the 2004-2017 monitoring period is available in Table S4 in the electronic supplement.

| Sensor | Sensor height (m) | |
| --- | --- | --- |
| | OPAWS | OHAWS |
| Snow depth | 2.4 | 2.2 |
| Air temperature/Humidity | 2.5 | 2.2 |
| Net radiation | N/A | 2.0 |
| Wind | 4.1 | 3.1 |
| Cumulative precipitation | 2.4 | 2.2 |
| Four component radiation | 2.7 | N/A |
