# Peer review of "Lake O'Hara alpine hydrological observatory: Hydrological and meteorological dataset, 2004-2017"

_Earth System Science Data, 2018_

## Referee Comment (RC1) · J. I. López-Moreno (Referee) · 22 Aug 2018

I have read with interest the manuscript about Lake O'Hara alpine hydrological observatory and I think that it is offered a very nice and very complete dataset for the Canadian Rockies. Data includes surface and groundwater hydrological and water quality, snow accumulation close to the annual peak and meteorological data from two stations. Such information can be very useful for validation of meteorological and hydrological models under complex topography and snow dominated basins. This is particularly valuable in the frame of the special issue where this article is expected to be allocated. The manuscript is properly written and organized and it contains the information necessary to understand the shared data, the quality control performed and potential uncertainties. The access to the data is very straightforward, reaching trough .zip folders to excel files with the different datasets. As the data is organized in excel with different sheets is not easy to be handled in an automatic way, but it should not be a big problem for users to prepare themselves the date for their own analyses.

Thus I recommend the publication of the manuscript with two minor comments:

- I think it would be useful to provide in the manuscript a table summarizing the information of the dataset woth the periods covered by each type of information. Despite the information is well explained it could help to have a quick idea about the different data offered. - Having a look to the datasets, my impression is that relative humidity data contains some inhomogeneities, 100% of humidity is only reached during short time in one of the stations while in the other is only during a relatively short period the 100% Rh is not reached. This should be checked and corrected or at least discussed in the manuscript.

Ignacio López
* * *

---

## Referee Comment (RC2) · Anonymous Referee #2 · 19 Oct 2018

Title: Lake O'Hara alpine hydrological observatory: Hydrological and meteorological dataset, 2004-2017.

Authors: Jesse He and Masaki Hayashi.

This manuscript introduces a comprehensive new dataset available for an alpine catchment in the Canadian Rockies. This kind of dataset are rare and therefore its relevance for hydrological studies in alpine catchments. In general the paper is well structured and properly written, however, I think there are things that need to be improved before accepted for publication. My main comment is the lack of details provided for the dataset and a more comprehensive examination of them. Data are easily accessible through the provided .csv files. I recommend this manuscript for publication after minor revision. Please find below more detailed comments.

General Comments:

A table summarizing the entire dataset, including gaps, recording frequency and available period is needed.

Refer to "temperature" as "Air temperature" throughout the manuscript. This will avoid confusions with water temperature.

I understand this is a "data manuscript" but, nevertheless, I would like to see a little bit more of data analysis. For example, more about inter-annual variability of the fluxes (e.g. precipitation, streamflow, air temperature), is there a year particularly interesting in the dataset that it is worth to look at in more detail, runoff ratios (easy to calculate and very informative) and mean hydrograph.

Provide coordinates for the weather and hydrometric stations (not found in the .csv files).

Specific comments:

L39: Do you mean "mean monthly air temperature"?

L78-79: If sensor height has changed this should be noted with the associated height and date, a table could be a good idea.

L94-97: The procedure used to fill precipitation gaps requires more details. For example, how much of the data this had to fill? And when exactly? This kind of details is very important for future hydrological applications. How much did the precipitation changed after wind undercatch correction (percentage)?

Figure 3: include coefficient of determination ($R^2$), the period used to generate this linear equation and you should force the line to pass through (0,0) to avoid negative precipitation events.

L121: For what years exactly water levels are not available?

L145-146: I would expect LiDAR to be much more accurate than any older and lower resolution DEM.

L159: Speculation, remove.

L167: Did you perform a statistical test to investigate runoff change? Your period of analysis is probably too short to do this.

Figure 4: Missing a legend.

---

## Author Comment (AC1) · 10 Dec 2018

We thank both reviewers for their constructive feedback. Attached is a .zip folder which holds our responses to comments, revised and marked-up manuscript, and supplementary information to be included with the final version of the paper.

Please also note the supplement to this comment: https://www.earth-syst-sci-data-discuss.net/essd-2018-83/essd-2018-83-AC1-supplement.zip

2018.

---

## Author Response (AR1)

We thank the two reviewers for their constructive comments, which we address below. The referee comments are in an *italic font*, and our responses are in the regular font. The line numbers in our comments refer to those in the marked version of the revised manuscript.

Comment 1

*I have read with interest the manuscript about Lake O'Hara alpine hydrological observatory and I think that it is offered a very nice and very complete dataset for the Canadian Rockies. Data includes surface and groundwater hydrological and water quality, snow accumulation close to the annual peak and meteorological data from two stations. Such information can be very useful for validation of meteorological and hydrological models under complex topography and snow dominated basins. This is particularly valuable in the frame of the special issue where this article is expected to be allocated. The manuscript is properly written and organized and it contains the information necessary to understand the shared data, the quality control performed and potential uncertainties. The access to the data is very straightforward, reaching trough .zip folders to excel files with the different datasets. As the data is organized in excel with different sheets is not easy to be handled in an automatic way, but it should not be a big problem for users to prepare themselves the date for their own analyses. Thus I recommend the publication of the manuscript with two minor comments:*

*I think it would be useful to provide in the manuscript a table summarizing the information of the dataset with the periods covered by each type of information. Despite the information is well explained it could help to have a quick idea about the different data offered.*
A table listing all time series, measurement interval, and years of availability has been added to the supplement (Table S2) and referenced in the text (Lines 127 and 136).

*Having a look to the datasets, my impression is that relative humidity data contains some inhomogeneities, 100% of humidity is only reached during short time in one of the stations while in the other is only during a relatively short period the 100% Rh is not reached. This should be checked and corrected or at least discussed in the manuscript.*
The HMP45 relative humidity sensors have a 2% measurement error when calibrated in the field, which could account for some of this inhomogeneity. We calibrate these sensors yearly. It is possible that these calibrations could cause the year-to-year differences if done improperly. The text has been edited to mention this issue (Lines 87-89).

Comment 2:

*This manuscript introduces a comprehensive new dataset available for an alpine catchment in the Canadian Rockies. This kind of dataset are rare and therefore its relevance for hydrological studies in alpine catchments. In general the paper is well structured and properly written, however, I think there are things that need to be improved before accepted for publication. My main comment is the lack of details provided for the dataset and a more comprehensive examination of them. Data are easily accessible through the provided .csv files. I recommend this manuscript for publication after minor revision. Please find below more detailed comments.*

*General Comments:*

*A table summarizing the entire dataset, including gaps, recording frequency and available period is needed.*
A table listing all time series, measurement interval, and years of availability has been added to the supplement (Table S2) and referenced in the text (Lines 127 and 136).

*Refer to "temperature" as "Air temperature" throughout the manuscript. This will avoid confusions with water temperature.*
The text has been edited to make the distinction between air and water temperature clear (Lines 39 and 161).

*I understand this is a "data manuscript" but, nevertheless, I would like to see a little bit more of data analysis. For example, more about inter-annual variability of the fluxes (e.g. precipitation, streamflow, air temperature), is there a year particularly interesting in the dataset that it is worth to look at in more detail, runoff ratios (easy to calculate and very informative) and mean hydrograph.*
We have expanded upon our analysis of runoff at the Lake O'Hara outlet and precipitation. In addition to the comparison of total discharge and precipitation, we have added the comparison of precipitation and late-summer flow, which represents the contribution of groundwater discharge (Lines 172-176). There was no correlation between the two variables, indicating that the groundwater discharge is likely controlled by the storage capacity and transmission characteristics of alpine aquifers. This has a major implication in our understanding of groundwater processes, but we refrained from presenting lengthy discussion of the importance of this finding.

*Provide coordinates for the weather and hydrometric stations (not found in the .csv files).*
We have added weather and gauging station coordinates to the supplement (Table S3) and to the metadata included within the dataset.

*Specific comments:*

*L39: Do you mean "mean monthly air temperature"?*
Yes, we have changed 'yearly' to 'monthly' (Line 39).

*L78-79: If sensor height has changed this should be noted with the associated height and date, a table could be a good idea.*
A record of sensor height measurements is provided with the supplement (Table S4). The Table 1 caption has been edited to refer to this record more clearly.

*L94-97: The procedure used to fill precipitation gaps requires more details. For example, how much of the data this had to fill? And when exactly? This kind of details is very important for future hydrological applications. How much did the precipitation changed after wind undercatch correction (percentage)?*
We have added a table in the supplement listing the date ranges of when precipitation was approximated and total precipitation approximated over each range (Table S1). After the wind undercatch correction was applied, precipitation increased by 8.6% and 0.3% at OPAWS and OHAWS respectively (Line 94).

*Figure 3: include coefficient of determination (R2), the period used to generate this linear equation and you should force the line to pass through (0,0) to avoid negative precipitation events.*
We have edited the figure to display the linear equation forced through the origin and the equation and coefficient of determination. We have rewritten the caption to describe which data are plotted in greater detail.

*L121: For what years exactly water levels are not available?*
Years missing corrected water level data are 2004 – 2008 at Lake O'Hara, 2004 – 2008 and 2012 at Opabin Lake. This is shown now in Table S2 of the supplement.

*L145-146: I would expect LiDAR to be much more accurate than any older and lower resolution DEM.*
We agree. We suspect that the difference is likely due to the inconsistency in the elevation datum used during our LiDAR survey compared to the older map. We have revised the sentence to make this point more explicit (Lines 149-151).

*L159: Speculation, remove.*
This is widely reported in the literature. We have kept the statement, but added a reference for support (Line 164).

*L167: Did you perform a statistical test to investigate runoff change? Your period of analysis is probably too short to do this.*
We agree. We have deleted the sentence on the trend in runoff (Line 176). For the correlation analysis between precipitation and discharge in this paragraph, we have included the statistical level of significance (p) (Lines 171-172).

*Figure 4: Missing a legend.*
We believe this comment refers to Figure 4a and 4b showing the annual and winter time series. We indicate these by annotating the lines instead of including a legend box, which would clutter the figures.

[revised manuscript text omitted]